# Method Development for Enteric Virus Recovery from Primary Sludge

**DOI:** 10.3390/v13030440

**Published:** 2021-03-09

**Authors:** Yarrow S. Linden, Christine S. Fagnant-Sperati, Alexandra L. Kossik, Joanna Ciol Harrison, Nicola K. Beck, David S. Boyle, John Scott Meschke

**Affiliations:** 1Department of Environmental and Occupational Health Sciences, University of Washington, 4225 Roosevelt Way NE, Suite 100, Seattle, WA 98195, USA; yarrow.linden@gmail.com (Y.S.L.); cfagnant@uw.edu (C.S.F.-S.); kossia@uw.edu (A.L.K.); joannacharrison@comcast.net (J.C.H.); nkj2@uw.edu (N.K.B.); 2PATH, 2201 Westlake Ave, Suite 200, Seattle, WA 98121, USA; dboyle@path.org

**Keywords:** poliovirus, enteric viruses, environmental surveillance, disease surveillance, sludge, environmental monitoring

## Abstract

Enteric viruses, such as poliovirus, are a leading cause of gastroenteritis, which causes 2–3 million deaths annually. Environmental surveillance of wastewater supplements clinical surveillance for monitoring enteric virus circulation. However, while many environmental surveillance methods require liquid samples, some at-risk locations utilize pit latrines with waste characterized by high solids content. This study’s objective was to develop and evaluate enteric virus concentration protocols for high solids content samples. Two existing protocols were modified and tested using poliovirus type 1 (PV1) seeded into primary sludge. Method 1 (M1) utilized acid adsorption, followed by 2 or 3 elutions (glycine/sodium chloride and/or threonine/sodium chloride), and skimmed milk flocculation. Method 2 (M2) began with centrifugation. The liquid fraction was filtered through a ViroCap filter and eluted (beef extract/glycine). The solid fraction was eluted (beef extract/disodium hydrogen phosphate/citric acid) and concentrated by skimmed milk flocculation. Recovery was enumerated by plaque assay. M1 yielded higher PV1 recovery than M2, though this result was not statistically significant (26.1% and 15.9%, respectively). M1 was further optimized, resulting in significantly greater PV1 recovery when compared to the original protocol (*p* < 0.05). This method can be used to improve understanding of enteric virus presence in communities without liquid waste streams.

## 1. Introduction

Enteric viruses are a leading cause of gastroenteritis, which is responsible for 2 to 3 million deaths per year globally [1,2,3,4]. Understanding the prevalence of enteric viruses within a community is critical for estimating disease burdens, tracking spread of pathogens, identifying outbreaks, and planning vaccine activities [5,6]. While clinical surveillance captures data on symptomatic individuals who access healthcare, environmental surveillance is an important supplement to clinical surveillance for monitoring sub-clinical or silent virus circulation via asymptomatic shedders, and confirming presence of vaccine-related viruses following vaccine campaigns [7,8,9]. Since enteric viruses are shed in stool, environmental surveillance is conducted by testing sources impacted by human waste, including wastewater and wastewater-impacted surface waters [10,11]. Environmental surveillance is used globally for enteric viruses such as rotavirus, norovirus, human enterovirus, adenovirus, and poliovirus (PV) [12,13,14,15,16].

The environmental surveillance program for PV has been highly developed and coordinated by the Global Polio Laboratory Network [5,6]. PV is an enterovirus that causes poliomyelitis, which affects the central nervous system and can result in acute flaccid paralysis causing lifelong disability and in some cases death [17]. While there has been over a 99% decline in global incidence of PV since the Global Polio Eradication Initiative was formed in 1988, regions at risk of PV exposure still exist [5]. Afghanistan and Pakistan have endemic PV transmission, and other countries are at risk of wild PV importation and vaccine-derived PV [7]. Due to its importance to eradication, environmental surveillance of PV is extensive, with over 2500 samples collected from endemic countries in 2017 [5,6]. The World Health Organization standard method for PV environmental surveillance concentrates 500 mL by aqueous polymer two-phase separation [8]. Three other methods are used, including (1) the bag-mediated filtration system, which concentrates 3–6 L samples by gravity filtration on a positively charged filter followed by skimmed-milk flocculation; (2) mixed cellulose ester filtration, which concentrates 0.5–1 L samples on a negatively charged filter after MgCl2 addition and acidification; and (3) ultrafiltration, which concentrates 1 L samples using Amicon ultrafiltration membranes [11,13,18,19,20,21,22,23]. However, all of these methods require liquid samples for concentration.

Some locations at risk for enteric virus or PV outbreaks do not have water-based sanitation systems, and instead rely on pit latrine waste systems. Over 17% of the global population uses pit latrines for sanitation, including 26% of rural and 10% of urban households [24]. A survey in Nigeria reported that 44.2% of households use pit latrines as their main sanitation system [25]. In Pakistan, 11% of rural households use pit latrines, and 40% do not have access to improved sanitation [26]. In urban areas of Pakistan, only 3% of households use pit latrines, while 26% lack access to improved sanitation [24]. In these areas of poor infrastructure with no wastewater collection system, shared community pit latrine samples become a primary source for environmental surveillance. Latrine waste systems are characterized by a higher solids content than typical wastewater (30% and 0.5% for latrine waste and wastewater, respectively). Total solids (TS) and total suspended solids (TSS) concentrations in pit latrine waste have been estimated to vary widely, with ranges between 1.2–54% TS and 2–19% TSS [27,28]. Primary wastewater sludge is characterized by a similarly high solids content, with a TS concentration ranging from 5–9% and an average of 6% [29]. Due to the high solids content in pit latrine waste and primary sludge, typical PV and enteric virus environmental surveillance methods are not adapted for these matrices [30].

The objective of this study was to develop and evaluate a protocol for enteric virus concentration and recovery for use with pit latrine waste and primary sludge. The objective was carried out by modifying two existing protocols for virus or bacteriophage recovery from varying media [31,32]. The first protocol initially evaluated hepatitis A virus recovery from oysters; this protocol was selected due to the relatively high recovery (46%) from a complex and dense sample matrix [31]. The second protocol had evaluated recovery of somatic coliphages from sewage sludge; this was evaluated due to the authors’ successful optimization of bacteriophage recovery from the same sample matrix evaluated in this study [32]. Experiments in this study were conducted using vaccine strain PV type 1 (PV1) and primary sludge. The two methods were tested and compared for PV recovery, and the method with higher PV1 recovery, lower limit of detection, and simpler usability was further optimized.

## 2. Materials and Methods

### 2.1. Study Organisms and Enumeration

Stocks of PV1 were prepared by growth on buffalo green monkey kidney (BGMK) cells and purified by chloroform extraction for storage at −80 °C [33]. PV1 was enumerated using a virus plaque assay. For the virus plaque assay, BGMK cells grown in 9.5 cm^2^ well plates were infected with 200 µL of the purified sample, overlaid with an Avicel RC-581 (FMC Corporation, Philadelphia, PA, USA) mixture [34,35], and incubated for 48 h (37 °C, 5% CO_2_) prior to staining. Relevant sample dilutions were plated in triplicate. Negative controls of phosphate-buffered saline (PBS) and unseeded sludge, and positive controls of PV1 titer were also plated. BGMK cells were obtained from Dr. Daniel Dahling (United States Environmental Protection Agency).

### 2.2. Samples

Primary sludge was collected on a single day from a local wastewater treatment plant in Seattle, Washington (USA), mixed thoroughly in the laboratory, aliquoted into 100 mL quantities, and frozen at −20 °C. These frozen sludge samples formed the matrix used for all seeded experiments, and had an average TSS of 26.11 g/L. For seeded experiments, samples were prepared by thawing overnight at 4 °C and then holding at room temperature for 2 h. Samples were then seeded with 10^4^ or 10^5^ plaque-forming units (PFU) of PV1, which had been de-aggregated via serial membrane filtration [18]. After seeding, samples were homogenized by vortexing (5 min) and then shaken (10 min) at room temperature with 50 mL of deionized water then added to improve processing. A Barnstead Lab-Line A Class MaxQ 2000 shaker was used for all shaking throughout the methods.

### 2.3. Preliminary Investigations

#### 2.3.1. Method 1

The prepared sample was processed by Method 1 (M1) as shown in Figure 1a and adapted from Mullendore et al. [31]. First, acid adsorption was performed by adjusting the sample pH to 4.8–5.0 using hydrochloric acid and then shaking (15 min, 200 RPM). The sample was centrifuged (20 min, 2000× *g*, 4 °C) and the supernatant discarded. The pellet was eluted in 100 mL of 0.05 M glycine (TCI, Tokyo, Japan), 0.14 M sodium chloride (pH 7.5) (NaCl, Fisher Scientific, Hampton, NH, USA) solution, by shaking (15 min, 200 RPM). The sample was centrifuged (20 min, 5000× *g*, 4 °C) and the supernatant saved. The pellet was re-eluted in 100 mL of 0.5 M threonine (VWR International, Radnor, PA, USA), 0.14 M NaCl (pH 7.5) solution by shaking (15 min, 200 RPM). The sample was centrifuged (20 min, 5000× *g*, 4 °C). The supernatant was collected and combined with the previously saved supernatant, while the pellet was discarded.

Secondary concentration was performed on the sample by skimmed-milk flocculation through addition of 1 mL of 5% (*w*/*v*) pre-flocculated skimmed-milk solution (Oxoid, Ltd., Hants, UK) per 100 mL sample. The sample was adjusted to pH 3.5–4.0, shaken (2 h, 200 RPM, room temperature) to allow floc formation, and centrifuged (30 min, 3500× *g*, 4 °C). The supernatant was discarded, and the pellet resuspended in 10 mL PBS (pH 7.4). The sample was stored overnight at 4 °C. The following day, the sample was purified by adding 5 mL Vertrel XF (E. I. du Pont de Nemours and Company, Wilmington, DE, USA), vortexing (2 min), and centrifuging (15 min, 3000× *g*, 4 °C). The aqueous phase (PBS) was collected. The remaining Vertrel fraction was re-extracted with the further addition of 5 mL 0.5 M threonine, 0.14 M NaCl, pH 7.5 solution, vortexing (2 min), and centrifuging (15 min, 3000× *g*, 4 °C). The aqueous phase (threonine) was combined with the PBS from the first extraction. A final secondary concentration by skimmed-milk flocculation was performed as described previously on the combined aqueous phases. The purified concentrated sample was stored at 4 °C until plaque assayed within 24 h.

#### 2.3.2. Method 2

The prepared sample was processed by Method 2 (M2) as shown in Figure 1b and adapted from Murthi et al. [32]. First, the sample was centrifuged (15 min, 6700× *g*, 4 °C) to separate the liquid and solid portions for separate processing (Figure 1b). The liquid fraction was passed through a positively charged 47-mm flat disc ViroCap filter (Scientific Methods, Granger, IN, USA) via membrane filtration. The filter was eluted in a 50 mL conical with addition of 10 mL eluate (1.5% beef extract (Becton, Dickinson and Company, Franklin Lakes, NJ, USA), 0.05 M glycine, pH 9.5 solution) and 4.5 g sterile glass beads (3 mm diameter), followed by vortexing (5–10 min). The sample was purified with addition of 5 mL Vertrel as described previously and stored at 4 °C for plaque assay within 24 h.

The solid fraction was eluted with 100 mL of 10% beef extract, 0.05 M disodium hydrogen phosphate (Millipore Corp, Bedford, MA, USA), 0.00625 M citric acid, pH 7.0 solution, by shaking (30 min, 350 RPM). The sample was then sonicated (3 min, on ice), followed by centrifugation (15 min, 6700× *g*, 4 °C). The pellet was discarded, while the supernatant was secondary concentrated by skimmed-milk flocculation and extracted with 5 mL of Vertrel, as described previously. The final sample was stored at 4 °C for plaque assay within 24 h.

### 2.4. Method 1 Optimization

M1 was optimized for PV1 recovery due to its higher initial PV1 recovery, lower limit of detection, lower time requirements, and simpler mechanics. Individual recoveries at each elution and extraction step were determined, and acid adsorption effects were examined. For all optimizations, M1 was performed as described previously with the following modifications: PV1 was spiked at 10^5^ PFU and the sample was extracted via Vertrel/PBS and Vertrel/threonine prior to overnight storage at 4 °C, eliminating the final skimmed milk flocculation and overnight storage steps.

#### 2.4.1. Solids Recovery

To determine the incremental benefits of a third elution on PV1 recovery during elution optimization, the PV1 recoveries from the glycine/NaCl elution, threonine elution, and an additional threonine elution were determined. The sample was processed through the acid adsorption step (Figure 1a). The sample was then eluted three times, and individual eluates were secondary concentrated via skimmed milk flocculation, extracted once via Vertrel/PBS (Vertrel/threonine re-extraction was eliminated), and assayed via BGMK plaque assay separately.

For extraction optimization, the PV1 recoveries from the Vertrel/PBS extraction and the Vertrel/threonine extraction were determined. The sample was processed through the first skimmed milk flocculation (Figure 1a). The sample was then extracted twice (Vertrel/PBS and Vertrel/threonine) and each extraction was assayed separately via BGMK plaque assay.

#### 2.4.2. Liquid Recovery

The inclusion and omission of acidification to pH 4.8–5.0 on PV1 recovery was determined. The prepared sample (acidified or not) was centrifuged to separate liquid from solids. The supernatant was concentrated via skimmed-milk flocculation, purified via Vertrel/PBS and Vertrel/threonine extractions, and assayed via BGMK plaque assay. The solid fraction was eluted (glycine/NaCl and threonine), concentrated (skimmed milk flocculation), purified (Vertrel/PBS and Vertrel/threonine), and assayed as described previously.

The effect of recombining the liquid and solid fractions prior to skimmed milk flocculation was determined. The prepared sample (acidified or not) was centrifuged to separate liquid from solids. The solid fraction was eluted twice (glycine/NaCl and threonine) and then combined with the liquid fraction. Skimmed-milk flocculation was performed on this combined sample, followed by purification (Vertrel/PBS and Vertrel/threonine) and assay via BGMK plaque assay.

### 2.5. Statistical Analyses

Statistical analyses were conducted using Microsoft Excel 2016. The 95% confidence intervals (CI) on PV1 recovery were calculated to determine the margin of error. Data sets were evaluated for normality by the Shapiro–Wilk test, and variance was assessed by the one-way analysis of variance (ANOVA) and F tests. Unpaired Student’s *t*-tests were used to compare recoveries between methods with normally distributed sample sets. The Kruskal–Wallis H test was used to determine if there was a difference in PV1 recovery between sample elutions during M1 optimization. The Nemenyi test was used for post hoc evaluation of which group pairings were different.

The limit of detection (PFU_1_), or number of PFU’s PV1 in the original sample, was calculated. The PFU’s in the final sample volume (PFU_2_) is a product of PFU_1_ and the percent recovery rate (R) (Equation (1)).
(1)$$PFU_{2}=PFU_{1}\times R$$

The PV1 concentration in the final concentrate (C_2_) is then calculated with the final concentrate volume (V_2_) (Equation (2)).
(2)$$C_{2}=\frac{PFU_{2}}{V_{2}}$$

The PFU’s entering tissue culture (PFU_TC_) is a product of the C_2_ and the concentrate volume entering tissue culture (V_TC_) (Equation (3)).
(3)$$PFU_{TC}=C_{2}\times V_{TC}$$

C_2_ can then be re-written (Equation (4)), and PFU_1_ is solved for (Equation (5)).
(4)$$PFU_{TC}=\frac{PFU_{1}\times R}{V_{2}}\times V_{TC}$$
(5)$$PFU_{1}=\frac{PFU_{TC}\times V_{2}}{R\times V_{TC}}$$

It was assumed 1 PFU PV1 was needed to enter the tissue culture plaque assay for detection (PFU_TC_).

The data presented in this study are available in Appendix A.

## 3. Results

### 3.1. Preliminary Investigations

M1 (26.1 ± 10.3%, 95% CI) yielded a higher mean PV1 recovery than M2 (15.9 ± 7.8%, 95% CI), though results were not statistically different (*p* = 0.145). The PV1 recoveries obtained with M1 ranged from 13.1 to 46.5% with a median of 22.0% (*n* = 8), while PV1 recoveries from M2 ranged from 11.9 to 23.0% with a median of 14.3% (*n* = 4). In M2, the PV1 recovery in the solids and liquid fractions were determined separately, resulting in a mean recovery of 8.4 ± 4.4% and 7.5% ± 5.5%, respectively (95% CI, *n* = 4). These recoveries were not statistically different (*p* = 0.68). Assuming a 0.2 mL assay volume in tissue culture, the limit of detection for PV in the original 100 mL sample for M1 and M2 is 192 PFU and 315 PFU, respectively.

### 3.2. Method 1 Optimization

#### 3.2.1. Solids Recovery

Elution optimization showed the fractional contribution of each elution step to the total mean PV1 recovery (Figure 2). Elutions 1 and 2 contributed the most to the total PV1 recovery with a mean recovery of 23.1 ± 13.8% and 21.0 ± 7.5%, respectively while Elution 3 contributed the least with a mean recovery of 4.3 ± 2.0% (95% CI, *n* = 8, Figure 2 and Figure 3). Elutions 1 and 2 resulted in significantly greater PV1 recovery compared to Elution 3 (*p* = 0.003 and 0.003, respectively; Kruskal–Wallis H test with Nemenyi). The total mean combined recovery for these elutions was 48.4 ± 20.6% (95% CI, *n* = 8, Figure 3), resulting in significantly greater total PV1 recovery when compared to the preliminary M1 protocol (*p* = 0.044).

Extraction optimization showed the fractional contribution of the Vertrel/PBS and Vertrel/threonine extractions on the total PV1 recovery (Figure 2). Vertrel/PBS extraction resulted in significantly greater PV1 recovery (37.9 ± 15.5%, 95% CI) compared to re-extraction in Vertrel/threonine (10.3 ± 4.5%, 95% CI) (*n* = 7, *p* = 0.004; Figure 4). The total mean combined recovery for these extractions was 48.2 ± 19.6% (95% CI, *n* = 7), resulting in significantly greater total PV1 recovery when compared to the preliminary M1 protocol (*p* = 0.038).

#### 3.2.2. Liquid Recovery

For combined samples (liquid fraction and pellet eluate pooled prior to skimmed-milk flocculation), the mean PV1 recovery was 32.8 ± 22.0% (acidified) and 44.8 ± 6.1% (non-acidified) (95% CI, *n* = 4, Figure 5). For the acid adsorption samples, the optimized recovery of the pellet alone was 48.2 ± 19.6% (95% CI, *n* = 7), and for the non-acidified samples, the optimized recovery of the liquid alone was 42.7 ± 9.9% (95% CI, *n* = 6).

When processing and assaying the liquid fraction and pellet separately, acidification affected the mean PV1 recoveries. Acidification resulted in a significant increase in the recovery from the pellet (48.2 ± 19.6% vs. 23.9 ± 7.3%; 95% CI, *p* = 0.023) and a decrease in the recovery from the liquid fraction (12.4 ± 8.3% vs. 42.7 ± 9.9%; 95% CI), though due to low sample numbers for statistical tests were unable to be conducted on this latter comparison (Figure 5). For these samples, the total mean PV1 recovery was 60.6 ± 19.6% for acidified and 66.6 ± 14.9% for non-acidified samples (95% CI). Processing the liquid fraction and pellet separately resulted in significantly greater PV1 recovery when compared to the combined samples for acidified samples (*p* = 0.012) and non-acidified samples (*p* = 0.028). Assuming a 0.2 mL assay volume in tissue culture and assaying the liquid and pellet separately, the limit of detection for PV in the original 100 mL sample for optimized acidified and non-acidified samples is 124 PFU and 113 PFU, respectively.

## 4. Discussion

### 4.1. Preliminary Investigations

Several factors likely contributed to the improved PV recovery of M1 over M2 in the preliminary investigations, including the acid adsorption step and eluent composition. Acid adsorption results in adhesion of virus particles to solid particulates via charge interactions [36]. These virus particles can then be concentrated by centrifugation and eluted from the solids [31]. The eluent composition also varied between the two methods. M1 used glycine/NaCl and threonine eluents, while M2 used beef extract. Amino acid eluents such as glycine and threonine have been shown to have similar or improved enterovirus and bacteriophage recovery from solid particles, compared to complex eluents such as beef extract, likely contributing to the higher PV1 recoveries using M1 [37,38,39]. Additionally, threonine results in lower PCR inhibition compared to beef extract [39].

In addition to the higher PV1 recovery and lower limit of detection with M1 compared to M2, M1 did not require processing the liquid portion via membrane filtration through a ViroCap disc filter, which resulted in logistical challenges. Positively charged ViroCap filters have been used for environmental surveillance of viruses in wastewater, as viruses adsorb to the filter material through charge interactions [13,40]. Viruses can later be eluted for analysis using a beef extract and glycine solution. Filtration of the liquid fraction (typically 100 mL) required up to 8 h, and up to 50 mL remained unfiltered due to excessive caking of solids on the filter. A previous study using 2” ViroCap cartridge filters resulted in a PV1 recovery from influent wastewater of 33.1% [19], compared to the 7.5% PV1 recovery from the liquid portion using ViroCap disc filters in this study. This difference in recovery may be due to the smaller available surface area on the disc filter, and modifications to the elution protocol to accommodate the disc filter’s geometry.

### 4.2. Method 1 Optimization

M1 was optimized for PV1 recovery from solids by determining individual recoveries at each elution and extraction step. The elution optimization showed that the inclusion of the second elution increased PV1 recovery by 21.0%, doubling the overall recovery (Figure 2a and Figure 3). However, a third elution step with threonine resulted in minimal additional recovery (4.3%) and, therefore, was not a necessary addition to the protocol. This differs from Mullendore et al., who found addition of a second and third threonine elution had similar fractional improvements on hepatitis A virus recovery from sedimented oyster tissue solids, with each elution improving the recovery by an additional 50% [31]. The extraction optimization process showed that while the first Vertrel/PBS extraction contributed the most to the high PV1 recovery (37.9%), the contribution of the Vertrel/threonine extraction to PV1 recovery (10.3%) was beneficial. Therefore, both extractions were utilized in the final protocol.

The M1 solids recovery optimization included Vertrel/PBS extraction prior to overnight sample storage and eliminated the second skimmed-milk flocculation step from the preliminary M1 protocol. These modifications may partially explain the optimized method’s higher PV1 recovery compared to the preliminary M1 investigations. By purifying the sample via Vertrel/PBS extraction prior to overnight storage, bacteria and fungi were inactivated, thus increasing PV1 survival. Additionally, the potential for PV1 loss was further minimized by reducing the number of sample processing steps.

M1 was additionally optimized for PV1 recovery with the exclusion of the acid adsorption step and the inclusion of liquids processing. PV1 recovery was highest from the solids when acid adsorption was utilized, whereas it was highest from the liquid fraction when acid adsorption was not utilized. Additionally, there was no statistical difference in PV1 recovery from the pellet in acidified samples compared to the liquid fraction from non-acidified samples (*p* = 0.552). These results suggest that the acid adsorption step was effective at adhering PV1 to particulates, thereby transferring PV1 from the liquid portion to solids portion. Despite this, the total PV1 recovery (liquid and solids) was similar between acidified and non-acidified samples.

When the solids eluate and liquid fraction were combined prior to the skimmed-milk flocculation step, PV1 recovery was 46% or 33% lower than if these sample portions were processed separately, for acidified and non-acidified samples, respectively (Figure 5). It is possible this was due to the high particulate levels in the sample exhausting the available protein flocs in the skimmed-milk flocculation method. Typically, when larger sample volumes (>100 mL) have been concentrated using skimmed-milk flocculation, the samples contain low particulate levels [41,42,43]. Future research could include increasing the skimmed-milk concentration to yield additional available proteins, potentially improving the method’s capabilities.

Based on the methods tested and optimizations completed, enteric virus recovery from primary sludge can be maximized through a relatively simple method with the exclusion of an acid adsorption step and by processing the solid and liquid fractions separately (Figure 6). After collection, samples should be kept at 4 °C, and processing should begin within 24 h. The solids should be eluted twice, secondary concentrated using skimmed-milk flocculation, and extracted with Vertrel/PBS and Vertrel/threonine. The liquid fraction should be concentrated by skimmed-milk flocculation, and extracted with Vertrel/PBS and Vertrel/threonine. The solids and liquids concentrates can be combined at this step and assayed or assayed separately. This would yield a 66.6% percent recovery (Figure 5), with a limit of detection 113 PFU/sample (1.13 PFU/mL).

If short breaks are required during processing, it is recommended these take place prior to extraction with Vertrel/PBS, or after the Vertrel/threonine extraction is complete. Overnight storage may take place after skimmed-milk flocculation, or after Vertrel/threonine extraction. Samples should be stored at 4 °C when not actively processed.

### 4.3. Limitations

Several limitations exist in this study. Due to equipment malfunction during the M1 solids optimization experiments, a portion of samples were stored overnight at 4 °C prior to Vertrel extraction, potentially contributing to variation in percent recovery. The WHO uses tissue culture when conducting environmental surveillance of poliovirus, as this confirms the detection of infectious virus. Due to this standard, tissue culture was used in this study design. However, tissue culture has inherent limitations: samples with lower virus concentrations or stressed viruses form plaques more slowly, and thus leave room for human error when interpreting plaque counts. Additionally, some viruses do not readily replicate in tissue culture, and so the assay used in this study may not be viable for surveillance of a full enteric virus community. It is anticipated these methods would be easily adaptable to molecular methods such as qPCR and/or RT-qPCR when using a nucleic acid extraction kit on the final supernatant that includes inhibitor removal, such as QIAamp PowerFecal Pro DNA Kit (QIAGEN, Hilden, Germany) or Quick-DNA/RNA Viral 96 Kit (Zymo Research, Irvine, CA, USA). Finally, as BGMK cells are not specific for poliovirus, it is possible that non-polio enteroviruses were amplified and detected. However, unseeded controls were plated for each experiment and experiments were seeded with PV1 at high levels to overcome baseline non-polio enterovirus levels.

Another limitation was that primary sludge was used as a surrogate for latrine waste. Latrine waste is typically characterized by a higher solids content, lower liquid content, and increased cellulose materials than primary sludge, although these vary by location [27,28,29]. While the same bulk sludge sample was aliquoted and frozen so each sample originated from the same source, it is possible there was variation within the bulk sludge sample that may have impacted PV1 recovery between samples. Additionally, experiments were conducted on different days, which could introduce variability. While PV1 is frequently used as a model for other enteric viruses, many factors (size, surface charge, shape, etc.) may impact virus recovery. Therefore, this method should be tested on additional viruses to determine its applicability.

## 5. Conclusions

This study demonstrated that 66.6% of seeded PV1 can be recovered from primary sludge by concentrating the liquid fraction using skimmed-milk flocculation and eluting the virus particles from solid particulates. Future work should fully evaluate the optimized method and examine the scalability of this method for processing larger volumes. As this method was tested with PV1, additional targets should be tested for recovery by tissue culture and molecular methods, including PV types 2 and 3, rotavirus, norovirus, and SARS-CoV-2. Finally, this method should be field-tested with latrine waste from a variety of locations to determine applicability for environmental surveillance of enteric pathogens.

## Figures and Tables

**Figure 1 viruses-13-00440-f001:**
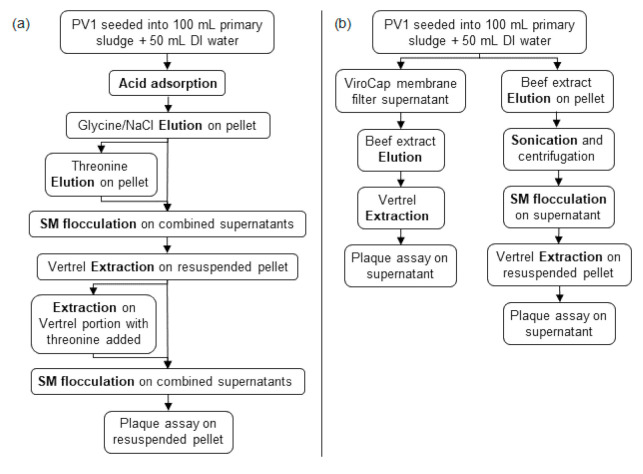
(**a**) Method 1 and (**b**) Method 2 experimental designs for preliminary investigations. PV1 is poliovirus type 1. DI is deionized. SM is skimmed milk flocculation.

**Figure 2 viruses-13-00440-f002:**
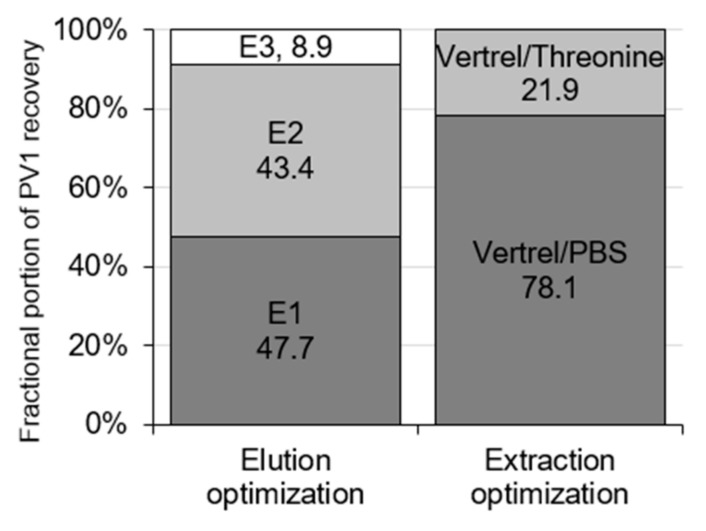
Percentage contribution of each elution to total mean recovery (48.4%) of poliovirus type 1 (PV1) from samples processed by Method 1 during the elution optimization and extraction optimization. Total contribution adds to 100%.

**Figure 3 viruses-13-00440-f003:**
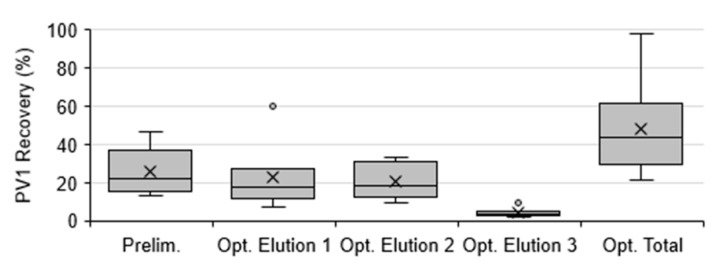
Recovery of poliovirus type 1 (PV1) from samples processed by Method 1 during preliminary (Prelim.) and elution optimization (Opt.) experiments. Box and whisker plot: lower, middle, and upper box lines show the first, second, and third quartiles, respectively; whiskers show the minimum and maximum data points; markers ‘×’ show the mean; and circles show the outliers.

**Figure 4 viruses-13-00440-f004:**
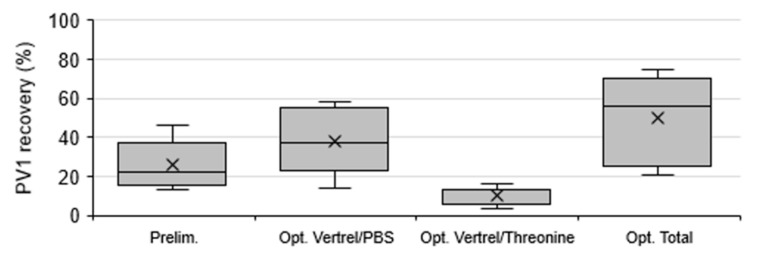
Recovery of poliovirus type 1 (PV1) from samples processed by Method 1 during preliminary (Prelim.) and extraction optimization (Opt.) experiments. Box and whisker plot: lower, middle, and upper box lines show the first, second, and third quartiles, respectively; whiskers show the minimum and maximum data points; and markers ‘×’ show the mean.

**Figure 5 viruses-13-00440-f005:**
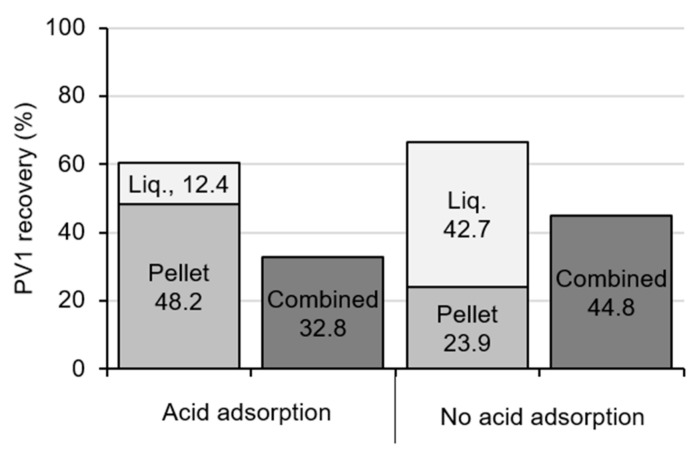
Mean recovery of poliovirus type 1 (PV1) from samples processed with or without acid adsorption.

**Figure 6 viruses-13-00440-f006:**
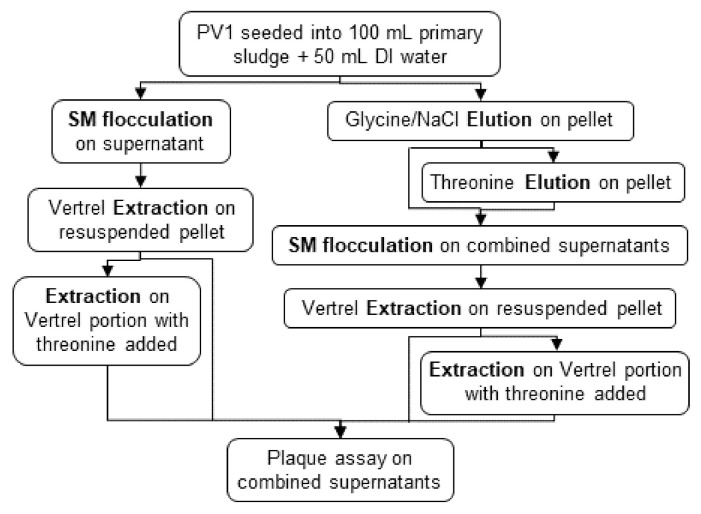
A flow diagram showing the processes for the optimized method for enteric virus recovery from primary sewage sludge.

## Data Availability

Data supporting reported results are available in Appendix A: PV Recovery Data.

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
