# Peer review of "Method Development for Enteric Virus Recovery from Primary Sludge"

_viruses, 2021, doi:10.3390/v13030440_

Round 1

Reviewer 1 Report

Optimization of environmental surveillance programs is critical as the authors suggest. The authors describe the surveillance for PV and enteric viruses. The introduction offers unique insight into global sanitation issues.

The two methods that are described are based on traditional principles.

Can the authors please address why PCR is not used alongside the plaque assay.

Can the authors also confirm that all these chemicals are available? The Vertrel is from duPont de Nemours company, how would one order from this company?

What might the limit of detection be for these assays?

If a researcher might need to leave the samples for any short period of time, is there a point where a break can occur and samples incubated at 4C perhaps, or must the assays be performed in order within time constraints?

How should samples be handled at point of collection and how long can they be stored prior to analysis by M1 or M2?

Reviewer 2 Report

The authors describe the results of their research to evaluate and optimize methods to recover poliovirus from primary sludge for environmental surveillance.

Overall, the manuscript was written well.  However, I do have some comments and suggest that the authors provide data of the combined method optimization results for comparison to the original method.

Line 52-57:  Suggest using different punctuation for the list of other methods used. As written, it is difficult to follow.

Line 78-79: Elaborate a little about why these 2 methods were chosen initially and rationale for modification.

Lind 81:  It is stated here that the “preferred” method was further optimized, but there isn’t an explanation of why a method is preferred.

Line 107:  Please list the piece of equipment used for shaking.  RPM is dependent on the instrument and cannot be repeated without this information.

Line 152: As stated earlier, explain why Method 1 (and not Method 2) was optimized.

Line 188:  Was normality and equal variance of data sets verified prior to running t-tests?

Line 191: I suggest including the 95% confidence intervals with the mean recovery efficiencies throughout the results section.

Lines 202-204: The recovery efficiencies reported in the text do not match the bars in Figure 2.

Line 204-205: Since 3 data sets were compared, an ANOVA seems more appropriate to assess the differences between the groups instead of pairwise t-tests.

Line 213:  Since the total PV1 recovered after elution 1-3 are compared to M1, it would be beneficial to plot M1 results to Figure 3.

Line 222:  It would also be beneficial to plot M1 results to Figure 4 for comparison.

Lines 304-307: PV1 recovery efficiency using the optimized parameters of M1 and compared to the results of M1 would have been a great conclusion of this work instead of the flow chart of the theorized method based on incremental analysis. It is highly recommended that these experiments be conducted, and results presented to complete this work.

Lines 318-320: what is meant by the assertion that cell culture analysis is challenging and leaves room for human error?

Reviewer 3 Report

The authors developed a recovery method of enteric viruses from primary sludge samples for detection by cell culture based on some modifications on the previously published methods. The virus recovery from sludge samples is challenging and this manuscript will give some knowledge in the field.

The reviewers and readers fully understand the importance of evaluating infectious viruses in sludge samples; however, for the purpose of understanding the prevalence of enteric viruses within a community (line 31), the molecular detection method can be one of the options as cell culture is not available or still difficult for many types of viruses. This should be discussed in the last paragraph in “4.3. Limitations”.

Specific comments

Line 195-196. The authors reported median values here and then mean recovery rates below. Why?

Line 215 & 224. Explanation of symbols and box plots should be described in each figure legend.

Line 229. the ‘difference’ was not statistically significant.

Line 295. How much were the PV1 recovery when these portions were combined or processed separately. If the difference was not statistically significant, they can be combined before the skimmed-milk flocculation step to optimize the method more simply.

Line 307. How much the overall PV1 recovery for the optimized method (Fig. 6)?

Line 332. Can the virus concentrate prepared by the optimized method (Fig. 6) be easily applied to molecular detection methods? Do you have some recommendations when applying to nucleic acid extraction and (RT-)PCR?

Line 334-337. Delete the paragraph.

Line 339. The authors should summarize the results obtained in this study with concrete values as they evaluated the PV1 recoveries.

Author Response

Thank you to the editor and the reviewers for your time.  Your comments have helped greatly to improve the manuscript.

Reviewer #3:

The authors developed a recovery method of enteric viruses from primary sludge samples for detection by cell culture based on some modifications on the previously published methods. The virus recovery from sludge samples is challenging and this manuscript will give some knowledge in the field.

The reviewers and readers fully understand the importance of evaluating infectious viruses in sludge samples; however, for the purpose of understanding the prevalence of enteric viruses within a community (line 31), the molecular detection method can be one of the options as cell culture is not available or still difficult for many types of viruses. This should be discussed in the last paragraph in “4.3. Limitations”.

Response: Thank you for bringing up this consideration.  This has been addressed in the limitations paragraph (lines 380-382): “Additionally, some viruses do not readily replicate in tissue culture, and so the assay used in this study may not be viable for surveillance of a full enteric virus community.”

It has also been added to the conclusions as a suggestion for future work (lines 405-407): “As this method was tested with PV1, additional targets should be tested for recovery by tissue culture and molecular methods, including PV types 2 and 3, rotavirus, norovirus, and SARS-CoV-2.”

Specific comments

Line 195-196. The authors reported median values here and then mean recovery rates below. Why?

Response: Thank you for noting this difference in reporting mechanisms. The mean and median results were reported in the preliminary methods to provide additional context to compare the methods, as sample numbers varied between the methods evaluated. In later evaluations, sample numbers were more similar and many comparisons were made. If adding median to each of these, the authors were concerned that the text would become too complex and the main message would be lost.

Line 215 & 224. Explanation of symbols and box plots should be described in each figure legend.

Response: Thank you for noting this omission. The following explanation has been added (lines 254-257 and 267-270): “Box and whisker plot: lower, middle, and upper box lines show the first, second, and third quartiles, respectively; whiskers show the minimum and maximum data points; markers ‘x’ show the mean; and circles show the outliers.”

Line 229. the ‘difference’ was not statistically significant.

Response: Thank you. Language indicating a comparison has been removed (lines 274-276): “For the acid adsorption samples, the optimized recovery of the pellet alone was 48.2±19.6% (95% CI, n=7), and for the non-acidified samples, the optimized recovery of the liquid alone was 42.7±9.9% (95% CI, n=6).”

Line 295. How much were the PV1 recovery when these portions were combined or processed separately. If the difference was not statistically significant, they can be combined before the skimmed-milk flocculation step to optimize the method more simply.

Response: Thank you for this suggestion. The authors agree that it would be simpler to combine the samples prior to skimmed-milk flocculation. However, PV1 recovery is significantly greater when samples are processed separately.  This has been clarified in the text (lines 346-348): “When the solids eluate and liquid fraction were combined prior to the skimmed-milk flocculation step, PV1 recovery was 46% or 33% lower than if these sample portions were processed separately, for acidified and non-acidified samples, respectively (Fig. 5).”

Line 307. How much the overall PV1 recovery for the optimized method (Fig. 6)?

Response: Thank you for the suggestion to clearly state this recovery. This has been added (lines 363-364): “This would yield a 66.6% percent recovery (Fig. 5), with a limit of detection 113 PFU/sample (1.13 PFU/mL).”

Line 332. Can the virus concentrate prepared by the optimized method (Fig. 6) be easily applied to molecular detection methods? Do you have some recommendations when applying to nucleic acid extraction and (RT-)PCR?

Response: Thank you, these questions have been addressed in lines 382-386: “It is anticipated these methods would be easily adaptable to molecular methods such as qPCR and/or RT-qPCR when using a nucleic acid extraction kit on the final supernatant that includes inhibitor removal, such as QIAamp PowerFecal Pro DNA Kit (QIAGEN, Hilden, Germany) or Quick-DNA/RNA Viral 96 Kit (Zymo Research, Irvine, CA, USA).”

Line 334-337. Delete the paragraph.

Response: Thank you for catching this inadvertent addition from the template.  This paragraph has been removed.

Line 339. The authors should summarize the results obtained in this study with concrete values as they evaluated the PV1 recoveries.

Response: Thank you for this suggestion. This addition will strengthen the manuscript’s conclusions paragraph. The percent recovery was added to the conclusions (lines 401-403): “This study demonstrated that 66.6% of seeded PV1 can be recovered from primary sludge by concentrating the liquid fraction using skimmed-milk flocculation and eluting the virus particles from solid particulates.”